# The Black Box Orchestra of Gut Bacteria and Bile Acids: Who Is the Conductor?

**DOI:** 10.3390/ijms24031816

**Published:** 2023-01-17

**Authors:** Soumia Majait, Max Nieuwdorp, Marleen Kemper, Maarten Soeters

**Affiliations:** 1Department of Pharmacy and Clinical Pharmacy, Amsterdam University Medical Center, 1105 AZ Amsterdam, The Netherlands; 2Department of Internal and Vascular Medicine, Amsterdam University Medical Center, 1105 AZ Amsterdam, The Netherlands; 3Department of Endocrinology and Metabolism, Amsterdam University Medical Center, 1105 AZ Amsterdam, The Netherlands

**Keywords:** gut microbiome, bile acids, type 2 diabetes mellitus, enterohepatic circulation, nutrition, probiotics, prebiotics

## Abstract

Over the past decades the potential role of the gut microbiome and bile acids in type 2 diabetes mellitus (T2DM) has been revealed, with a special reference to low bacterial alpha diversity. Certain bile acid effects on gut bacteria concern cytotoxicity, or in the case of the microbiome, bacteriotoxicity. Reciprocally, the gut microbiome plays a key role in regulating the bile acid pool by influencing the conversion and (de)conjugation of primary bile acids into secondary bile acids. Three main groups of bacterial enzymes responsible for the conversion of bile acids are bile salt hydrolases (BSHs), hydroxysteroid dehydrogenases (HSDHs) and enzymes encoded in the bile acid inducible (Bai) operon genes. Interventions such as probiotics, antibiotics and fecal microbiome transplantation can impact bile acids levels. Further evidence of the reciprocal interaction between gut microbiota and bile acids comes from a multitude of nutritional interventions including macronutrients, fibers, prebiotics, specific individual products or diets. Finally, anatomical changes after bariatric surgery are important because of their metabolic effects. The heterogeneity of studies, diseases, bacterial species and (epi)genetic influences such as nutrition may challenge establishing specific and detailed interventions that aim to tackle the gut microbiome and bile acids.

## 1. Introduction

The increasing trend of type 2 diabetes mellitus (T2DM) is a serious and growing challenge for public health and our health care system [1]. T2DM is associated with an increased risk of developing vascular complications, which are a leading cause of morbidity and mortality in diabetic patients. Western lifestyle including an inactive, sedentary behavior and high-caloric nutritional intake plays a central role in the pathogenesis of T2DM [2]. In this current review paper, we aim to provide a general view on the reciprocal interaction between the gut microbiome, bile acids and nutrition.

### 1.1. Gut Microbiome and Type 2 Diabetes Mellitus

Over the past decades the potential role of the gut microbiome in T2DM has been revealed. More than 10^14^ bacteria reside in the human gut [3]. Because of the development of new analytical methods, such as high-throughput metagenomic sequencing, a more comprehensive understanding of their function in relation to metabolic disorders has been established. Multiple studies have linked an imbalance in the gut microbiome with a variety of diseases, including diabetes and obesity [4,5,6]. In particular, low bacterial alpha diversity (microbiome diversity applicable to a single sample) is associated with the occurrence of obesity and T2DM [7]. The gut microbiome plays a crucial role in the metabolism of bile acids; these amphipathic acidic steroids on their turn have endocrine, metabolic and (anti-)inflammatory effects [8,9,10].

### 1.2. Bile Acid Metabolism

In the liver, biosynthetic pathways convert water-insoluble cholesterol into water-soluble molecules with detergent properties that are referred to as primary bile acids. Bile acids constitute a large family of molecules composed of a steroid structure with four rings and five or eight carbon side chains terminating carboxylic acid. Each bile acid has a specific number and orientation of hydroxyl groups (-OH) [8]. The principal primary bile acids in humans are cholic acid (CA) and chenodeoxycholic acid (CDCA). Ursodeoxycholic acid (UDCA) is the 7ß epimer of CDCA. Bile acids can be conjugated to either taurine or glycine (humans) to increase hydrophilicity [11]. Bile acids are preferably conjugated to taurine, which is reflected in the predominance of these conjugates in the murine bile acid pool [12]. Because taurine is less prevalent in the human diet, the human bile acid pool is predominantly glycine-conjugated [13,14]. In addition to solubility, conjugation is known to influence the toxicity of bile acids. In unconjugated form, bile acids appear to be more cytotoxic [8,15].

After ingestion of a meal, bile acids are released from the gallbladder into the small intestine to facilitate lipid emulsification, support metabolic endocrine processes and modulate host inflammatory responses. Specifically, and pertinent to this paper, bile acids are deconjugated and dehydroxylated by the gut bacteria. Intestinal bacteria can convert primary bile acids into secondary bile acids by removing a hydroxyl group. Thus, CA becomes deoxycholic acid (DCA) and CDCA becomes lithocholic acid (LCA) [15]. Efficient and active absorption of primary bile acids in the terminal ileum and passive transport of secondary bile acids in the colon, followed by hepatic uptake from the portal blood and re-secretion into bile, results in the presence of the bile acid pool within the human body [16].

### 1.3. Primary and Secondary Bile Acids: Role for the Microbiome

The cycling of bile acids between liver and the intestine is coined the enterohepatic circulation and has a frequency that is only partially determined by food and the pattern of food intake [17]. The enterohepatic circulation through liver, gallbladder, intestine and portal vein back to the liver is highly dynamic. Timing of (re)circulation is dependent on the chemical structure and conjugation status of bile acids [8]. Generally, conjugated bile acids do not pass the enterocyte membrane and are taken up by active transport more distally in the small intestine [18]. After absorption, the liver efficiently extracts most bile acids from the portal vein for re-secretion. The existence of such a circulating pool ensures the presence of adequate bile acid concentrations at the sites of physiological actions; particularly in the bile canaliculi to promote bile formation, in the gallbladder to prevent cholesterol crystallization, the intestinal lumen to facilitate the absorption of dietary fat and soluble vitamins, and endocrine–metabolic actions within the gut and pancreas which are mediated by the Takeda G coupled receptor (TGR) 5 and the Farnesoid X receptor (FXR) [10,16]. Absorbed bile acids form a postprandial peak in plasma up to a concentration of approximately 1–20 µmol/L 30–90 min after the meal [19]. Conjugation of bile acids increases their solubility, aiding secretion into bile. Although intestinal bacteria deconjugate bile acids, the majority of the human bile acid pool (~80%) reside in its conjugated form throughout the enterohepatic cycle [19]. Several Gram-positive bacteria, such as *Lactobaccili,* deconjugate primary bile acids [20]. The deconjugation steps are carried out by Gram-negative bacteria. Secondary bile acids are formed after oxidation and dihydroxylation and these processes are performed by Gram-positive *Clostridium* [21,22,23].

Around 2006, bile acids emerged as potential regulators of systemic energy homeostasis [24,25,26]. For example, binding of LCA and DCA to the G protein-coupled receptor TGR5 in the intestine strongly induces secretion of the incretin GLP-1, thereby affecting glucose metabolism. The association between human bile acids and glucose metabolism originated from the observation that patients with T2DM have increased bile acid pools [13]. In a study of bile acid kinetics from 2021, it was found that patients with T2DM have higher plasma levels of (conjugated) CDCA and (conjugated) DCA compared to control [14]. Another study showed similar increases in deoxycholic acid pool size in patients with T2DM [7]. However, some studies have demonstrated similar bile acid pools in subjects with and without T2DM. Variety within the T2DM group depends on the presence of insulin resistance [15], BMI [16] and presence of individual BA species [17].

As such, both the gut microbiome and bile acids show intriguing changes in T2DM patients. Therefore, with this review we aim to provide a better understanding of the reciprocal interaction between the gut microbiome and bile acids and their combined role in T2DM (see Figure 1). Changes in the gut microbiome can contribute to various homeostatic processes including shaping the bile acid composition with consequences for bile acid receptor activation. Moreover, we address the influence of nutritional intake and weight-loss (surgeries) on this interaction. We tried to rely on human studies, but did not exclude animal work where needed. In more detail, we aim to look at different factors that might disrupt the harmony between the gut microbiome and bile acids and their effect on T2DM.

## 2. Effects of Bile Acids on the Gut Microbiome

A few studies have demonstrated the effect of certain bile acids on the host gut microbiome composition which in general can be brought back to cytotoxicity, or in the case of the microbiome, bacteriotoxicity. However, studies that particularly dived in to bile acid effects (e.g., bile acid supplementation or sequestrants) are quite scarce.

### 2.1. Direct Effect of Bile Acids on the Microbiome

Although very old studies already proposed the inhibitory effect of unconjugated bile acids on bacteria, the mechanisms were not clarified at that time [27]. Islam et al. proposed that bile acids function as a host factor controlling the microbiome via their bactericidal action via membrane damage [28]. Moreover, the hydrophobicity of the bile acid corresponds with its affinity for the phospholipid bilayer of the bacterial cell membrane [29] In addition, Islam et al. suggest that DCA, which is a predominant bile acid in humans, applied a strong selective pressure capable of altering the microbiome composition in the cecum of rats [28].

Bile acids might pose antibacterial action throughout their amphipathic nature and the presence of a steroid nucleus [30]. This might be supported by their inhibitory character in bioenergetics processes by intracellular acidification, dissipation of the proton motive force, induction of DNA damage and protein denaturation [31,32,33]. In a study of Tian, it was shown that unconjugated bile acids are more potent to antibacterial activity compared to conjugated bile acids. One of the explanations is that unconjugated bile acids passively pass membranes, thereby causing intracellular toxicity [34]. In this regard, the study of Sannasiddappa et al. found that bile acids can kill *Staphylococcus aureus* in different ways depending on their concentrations and physicochemical properties. Notably, as *Staphylococcus aureus* is a bacterium that can cause deathly infections worldwide [35].

As stated above, unconjugated bile acids are more toxic compared to conjugated bile acids as the amphipathic nature of bile acid is responsible for the damage of the bacterial membrane [28]. Unfortunately, few studies have investigated the effect of bile acids therapy on bile acids in humans [8]. In a study of Ihunnah et al., rats were orally supplemented with cholic acid. A taxonomic change in their gut microbiome was observed with an increase in bile acid-metabolizing phyla and a decrease in bile acid-intolerant bacteria. Specifically, an increase of Firmicutes from 54% to 95% of the total microbiome was found and a decrease from 33% to less than 1% of Bacteroidetes and Actinobacteria was found [36]. These microbial changes were associated with an obesity-like bacterial community with an elevated energy-harvesting capacity [37] In addition, bacteria such as Lactobacillaceae produce bile salt hydrolase (BSH). BSH is involved in deconjugation of bile acids and plays a key role in bile acid metabolism. A possible explanation for the increased plasma levels of sulfated bile acids is the lower excretion of bile acids into the gallbladder [38].

The human studies that are available on bile acid-mediated effects on the microbiome are very scarce and concern UDCA supplementation. Here, UDCA treatment modestly influences the relative abundance of microbial species in stool. Indeed, gut microbial networks show an overrepresentation of *Faecalibacterium prausnitzii* after UDCA treatment and an inverse relationship between *Faecalibacterium prausnitzii* and *Ruminococcus gnavus* [39] Moreover, other studies that focused on UDCA in ulcerative colitis and intrahepatic cholestasis of pregnancy also showed modest changes in gut microbiota composition [40,41].

### 2.2. Effects of Bile Acid Withdrawal on the Microbiome

Alternatively, the absence of bile acids may have an effect on the gut microbiome. In this regard, the so-called bile acid sequestrants are used for multiple indications. Effectively, these drugs disrupt the enterohepatic cycle. In a murine nonalcoholic steatosis model, the addition of the bile acid sequestrant sevelamer, a phosphate-binding crosslinked polymer that also showed high affinity for bile acids, prevented hepatic steatosis and associated parameters [42]. Moreover, sevelamer not only lowered portal levels of total bile acids (inhibiting hepatic and intestinal FXR), but also improved a lower α-diversity and prevented decreases in *Lactobacillaceae* and *Clostridiaceae* as well as increases in *Desulfovibrionaceae* and *Enterobacteriaceae.* A partially comparable study was performed in patients with primary biliary cholangitis with cholestyramine (a bile acid-binding resin) [43] Although patient responses to treatment showed great interindividual variability, compositional shifts of the microbiome were observed that involved enrichment of two *Lachnospiraceae* species and *Klebsiella pneumonia*, a commensal bacterium. Notably, it cannot be excluded that bile acids binding to sequestrants have biological effects.

In short, alterations in bile acid composition with consequences on the gut microbiome in patients with T2DM haven been found [10,11,17,18,19,20]. Antibacterial characteristic and potent toxic implications of bile acids might affect gut microbial changes with consequences in T2DM [28]. Higher levels of conjugated bile acids are associated with an increased risk of T2DM [18]. Whether increased bile acids concentrations in the gut are also related to lower bacterial diversity via cytotoxicity is yet to be elucidated.

## 3. Effects of the Gut Microbiome on Bile Acids

Reciprocally, the gut microbiome plays a key role in regulating the bile acid pool by influencing the conversion and (de)conjugation of primary bile acids into secondary bile acids. Three main groups of bacterial enzymes responsible for the conversion of bile acids are bile salt hydrolases (BSHs), hydroxysteroid dehydrogenase (HSDHs) and enzymes encoded in the bile acid inducible (Bai) operon genes [44]. Also, pro-and antibiotics can alter bile acid levels and gut microbiome composition, suggesting its role in metabolic improvement [45].

### 3.1. Bacterial Bile Salt Hydrolases (BSH)

BSH enzymes are mainly present in Gram-positive bacteria firmicutes (*Lactobacillus* and *Enterococcus*) and in certain Gram-negative bacteria bacteroidetes [46]. BSHs catalyse the deconjugation of primary bile acids by hydrolysing the glycine or taurine group. The glycine or taurine group are further metabolised to carbon dioxide and ammonia [46]. Certain bacteria can use ammonia as an energy source [20]. De Smet et al. demonstrated that BSH activity is substrate-specific [47]. *Lactobacillus plantarum* preferably hydrolysed gDCA, whereas *Lactobacillus animalis* showed higher affinity for tDCA. Bacteroides were found to be responsible for the conversion of cholic acid to deoxycholic acid in mice, and *Clostridium scindens* tend to play a role in the conversion of UDCA to LCA in vitro [48,49].

### 3.2. Bacterial Hydroxysteroid Dehydrogenase (HSDH)

The oxidation and epimerisation of bile acids are catalysed by HSDHs [44]. These reactions need two distinct enzymes, either from the same or from different bacteria and occur in two steps. The first step is the oxidation of the hydroxyl group which is catalysed by 7α-HSDH. The second step is the reduction reaction carried out by 7β-HSDH [44]. For example, cholic acid can be epimerised to form derivates such as ursocholic acid or isocholic acid, and chenodeoxycholic acid can be epimerised to ursodeoxycholic acid or isochenodeoxycholic acid. In a study of MacDonald [50], *Eubacterium lentum* showed hydroxysteroid dehydrogenase activity responsible for the conversion to the 3-and 7-keto-derivatives [50].

### 3.3. Enzymes Encoded in the Bile Acid Inducible (Bai) Operon Genes

The third group of bacterial enzymes are enzymes encoded in the bile acid inducible (Bai) operon genes [44]. Bai operon genes are responsible for the dehydroxylation of primary bile acids resulting in the formation of DCA and LCA. In more detail, the operon is composed of 8 genes: BaiA2, BaiB, BaiCD, BaiE, BaiF. BaiN, BaiH and BaiO. All genes are responsible for a specific step in the dihydroxylation and present in many bacterial strains [44,51,52].

### 3.4. Effects of the Microbiome on Bile Acids: Probiotics

Apart from the bacterial capabilities to modulate bile acids, pro- and antibiotics will influence on the bacterial composition in the gut and the bile acid composition. Probiotics are living microorganisms with the purpose of beneficial health effects. As such, probiotics may support gut health and consist of species such as *Lactobacillus* and *Bifidobacterium.* Otherwise, it may be difficult to discern between probiotics and bacterial commensals. Because of the earlier identification of the relation of the commensal butyrogenic *Anaerobutyricum soehngenii* (that harbours genes encoding a bile acid (BA) sodium symporter and BA hydrolases) with improved insulin sensitivity, Koopen et al. investigated acute effects of infusion of this bacterium by nasogastric tube. *Anaerobutyricum soehngenii* tended to increase the postprandial excursions of secondary bile acids [53]. More interesting are different bile acids correlated with the GLP- 1, which had previously been shown for glycine- conjugated DCA [54]. Another study was performed with supplementation of *Lactobacillus reuteri* DSM 17938. Subjects receiving the highest dose of *Lactobacillus reuteri* displayed within-group increases in insulin sensitivity index (ISI) and serum levels of the secondary bile acid deoxycholic acid (DCA) compared with baseline with no effect after 12 weeks on HbA1c, liver steatosis, adiposity or microbiota composition [55] In contrast, *Lactobacillus casei* Shirota supplementation during 12 weeks had no effect on bile acids nor glucose homeostasis [56].

### 3.5. Effects of Gut Microbiome on Bile Acids: Antibiotics

Researchers have modulated the gut microbiome via the administration of antibiotics. Regarding this matter, Vrieze et al. studied 20 male obese subjects with metabolic syndrome who received either vancomycin or amoxicillin for 7 days in a single blinded randomised trial. Vancomycin decreased intestinal microbiota diversity (lower Gram-positive bacteria (mainly Firmicutes) and higher Gram-negative bacteria (mainly proteobacteria)), peripheral insulin sensitivity and bile acid dihydroxylation; thus, increasing postprandial plasma primary bile acids levels. These effects were not found after amoxicillin treatment [57]. Another interesting study, but slightly away from the topic of T2DM, was performed by Yukawa-Muto et al. Patients with hepatic encephalopathy were treated with the non-absorbable antibiotic drug rifaximin and showed differences between responders and non-responders with regard to the microbiome [58]. Remarkably, the sensitivity of *S. salivarius* and *R. gnavus* to RFX depended on conjugated secondary bile acid concentrations, although the exact mechanism of this bile acid–microbiome interaction was not elucidated.

### 3.6. Effects of Fecal Microbial Transplantation (FMT)

FMT has attracted great attention under multiple conditions including T2DM, gastrointestinal, hematologic, neurologic, metabolic, infectious, autoimmune disorders and infections, of which most notable are clostridium difficile infections [59]. Modulation of the gut microbiome by FMT is expected to have an effect on luminal bile acid metabolism. Within the treatment of clostridium difficile, which was unambiguously described in 2013, FTM has been widely used to treat patients. It seems that microbial bile salt hydrolases (see above) mediate the efficacy of FMT in the treatment of recurrent Clostridium difficile infection [60]. Some clostridium studies showed that fecal microbiota diversity of recipients significantly increased, (resembling the donor) with a reduction of primary bile acids and higher secondary bile acids [61]. Remarkably, it has been shown that the donor metabolic signature drives the effect of FMT on various aspects such as recipient insulin sensitivity and energy expenditure. Intriguingly, the transplantation of feces from metabolic syndrome subjects results in higher levels of lithocholic and deoxycholic acid comparable with T2DM subjects [62]. This was also seen in a study with encapsulated FMT in which subjects receiving donor FMT capsules had shifts in the microbiome and bile acids profile that resembled the donor [63].

### 3.7. Effects of Change in Bile Acids on FXR-Mediated Composition of the Bile Acid Pool

The intestinal flora influences the diversity of bile acids and thus bile acid synthesis and transportation. Changes in bile acid composition will also impact FXR activation [8]. CDCA, CA and DCA are agonists of FXR; UDCA is an antagonist. Therefore, a gut microbiome–bile acid–FXR axis exists [46]. Here FXR activation will have different consequences beyond bile acid biosynthesis. FXR also induces ileal bile acid-binding protein and organic solute transporter α/β to expel bile acids from the portal circulation. Active FXR transactivates the expression of the bile salt export pump to stimulate bile acid secretion from the liver to the biliary duct but inhibits the expression of the Na^+^-taurocholate co-transporting polypeptide and reduces bile acid reabsorption from the blood to the liver. These mechanisms are described in more detail elsewhere [46].

## 4. The Effects of Nutrition on the Gut Microbiome and Bile Acids

Further evidence of the reciprocal interaction between gut microbiota and bile acids indicates that the ingestion of food will have both consequences. A multitude of nutritional changes will exert changes on the microbiome and bile acids. First this is true for the macronutrients, but also specifically for fibres and other prebiotics. Specific individual products have modulating effects and this is also true for the way of conservation such as fermentation, although the latter has an effect on the microbial content of the food and could be regarded as probiotic. We discuss these items under the heading nutrition analogous to the prebiotics. Furthermore, anatomical changes such as bariatric surgery and parenteral nutrition are important because of their metabolic effects.

### 4.1. Macronutrients, Bile Acids and the Microbiome

It has been long known that the ingestion of a fatty meal increases the postprandial bile acid response the most. Sonne et al. have shown this very elegantly for normal glucose-tolerant people as well as for T2DM patients, although some degree of variation hampered significance at all timepoints and nutritional interventions [64]. Morton et al. also showed this increased bile acid response after lipid-rich meals. The explanation for increased bile acid excursions after lipid-rich meals lies most probably within increased gallbladder emptying, increased enterohepatic circulation of all bile acid species and possibly acute effects of macronutrient composition constituting an important regulator of postprandial bile acid pool composition [54,64,65]. A large study that exposed healthy young adults to a 40% high fat diet found lower alpha diversity, but the authors did consider the bile acids [66]. Wilson et al. interpreted and discussed key results of recently published studies on the effects of dietary change and nutritional intervention on the human microbiome from around the world. They point at the fact that modifications of dietary habits, e.g., the transition from rural to urban (high fat) dietary intake and vice versa, caused reciprocal changes in the gut microbiome and relevant shifts in bile acids. The effects of high fat diets in murine models (that contain more rodent, but not human insight per se) have been reviewed elsewhere [67]. However, the high fat diet consequences in rodents (natural carbohydrate eaters) are not fully incomparable to humans [68], although some consequences such as decreased alpha diversity and increased secondary bile acids occur in both species.

A low carbohydrate high protein diet (including healthy components) improved gut microbiome composition in individuals with chronic spinal cord injury, including increased numbers of bacteria implicated in fibre metabolism and reduced bacteria communities linked to cardiovascular disease [69].

### 4.2. Fibre, Prebiotic and Fermentation Effects on Bile Acids and the Microbiome

Dietary fibre contains fermentable carbohydrates; moreover, these fibres or indigestible carbohydrates can be fermented by the gut microbiome to short-chain fatty acids [70]. The three main SCFAs products are acetate, butyrate and propionate. A variety of gut bacteria are responsible for SCFA production. SCFAs play a role in immune regulation, intestinal epithelium integrity, insulin secretion and β-cell proliferation [71]. SCFAs seemed to exert a plasma cholesterol-lowering effect in rat fed sugar-beet fibre by inhibiting bile acid and cholesterol absorption or suppressing cholesterol synthesis [72]. Whereas fibers can be clearly described, the definition for prebiotics is somewhat more complicated and reviewed extensively elsewhere [73]. Prebiotics can be described as selectively fermented ingredients capable of inducing specific changes in the composition and/or activity of the gut microbiome, conferring benefit upon the health of the host thereby meeting three requirements: (a) resistance to gastric acidity and hydrolysis and absorption; (b) fermentation by intestinal bacteria; (c) selective stimulation of the growth/activity of intestinal bacteria associated with health and wellbeing [73].

A simple example of a prebiotic is the laxative lactulose that is perhaps most often used in constipation and hepatic encephalopathy. Comparable with acarbose, lactulose increases the amount bacteria *Lactobacillus* and *Bifidobacterium*, which coincides with reductions in secondary bile acid production [74]. The authors of this paper actually suggest that lactulose may have a role in the management of T2DM. Acarbose, which is also a non-digestible sugar, may have the same effect as lactulose [74]. Apparently, another sugar, inulin-type fructans, has a differential effect on the gut microbiome, modestly increasing *Bifidobacterium* without effects on the ratio between primary and secondary bile acids [75]. We are not aware of studies that specifically investigated the effect of more natural prebiotic compounds (i.e., unprocessed vegetables) on both the gut microbiome and bile acids. However, an example of a study investigating the effect of diets rich in fibre and plant food sources showed favorable gut microbiome composition in people consuming such diets [76].

### 4.3. Individual Products and Dietary Habits

Thompson showed that daily avocado consumption lowered fecal bile acid concentrations and increased relative abundances of bacteria capable of fibre fermentation, providing evidence that this nutrient-dense food affects digestive physiology, as well as the composition and metabolic functions of the intestinal microbiota [77]. Blueberries have comparable effects due to their high levels of anthocyanins, polyphenols known to influence the microbiome [78]. Blueberry consumption increases glycine-conjugated BAs and reduces secondary BA levels with significant changes in the bile acids. For the consumption of walnuts, more or less the same is valid: increasing relative abundances of Firmicutes species in butyrate-producing Clostridium clusters whilst reducing secondary bile acids [79].

A study by Jiang et al. investigated over 1800 adults and found that a so-called “fruit-microbiota index”, created from 31 identified microbial features, was positively associated with fruit intake and negatively with T2DM risk. Here, cholic acid associated negatively with the index and positively with the risk of T2DM [77]. The Mediterranean diet has long been advocated as a valuable strategy for healthy metabolism. Meslier et al. show that a Mediterranean diet intervention increases the fibre-degrading *Faecalibacterium prausnitzii* genes involved in microbial carbohydrate degradation linked to butyrate metabolism alongside decreased fecal bile acids [80]. A similar Mediterranean diet study showed a significant increase in Lachnospiraceae NK4A136 and decreased fecal bile acids (HCDA and cholate) [81]. A four day-only switch to the Mediterranean diet was less effective despite the fact that some changes were noticed [82].

### 4.4. Bariatric Surgery Effects on Bile Acids and Microbiome

Human studies have shown correlations between the gut microbiome and bile acids after bariatric surgery. Krajmalnik-Brown et al. used co-occurrence-network analysis that demonstrated interactions between fecal microbial phylotypes and bile acids after Roux-and Y gastric bypass (RYBG) surgery. Fusobacterium, Veillonella, Enterococcus, Akkermansia, and Streptococcus were substantially present in the feces and were negatively correlated with conjugated LCA, CDCA, CA and DCA. Fecal presence of Ruminococcus, Coprobacillus, Holdemania, Eggerthella, and Dorea were positively correlated with primary and secondary bile acids [83]. In a review of Liu the effect of bariatric surgery in patients with T2DM was discussed/explained. After weight-loss surgery both bile acid levels and the gut microbiome were altered and were associated with metabolic improvement in obesity, glucose tolerance and insulin sensitivity. This suggests a role of bariatric surgery in metabolic changes and microbiome–bile acid interaction [84]. However, bile acids generally increase after RYBG in contrast to dietary weight loss [85,86]. Still the increase of bile acids that coincides with higher FGF19 and GLP-1 levels suggests an anorectic effect of surgery. Human subjects consistently show an increase in gut microbiota diversity, spatial organization and stability, and specifically Proteobacteria after RYGB [85]. In contrast to RYGB, laparoscopic sleeve gastrectomy shows lower fecal bile acids and greater abundance of specific bacterial taxa and α-diversity that may contribute to the metabolic changes [87]. This was supported by a study of Kural et al. which showed that patients after surgery had lower BMI and higher bacterial diversity with an increased Firmicutes/Bacteroidetes ratio and lowered Prevotellaceae and Veillonellaceae [88]. Another interesting technique is the duodenal–jejunal bypass liner (DJBL), which leads to weight loss and restoration of insulin sensitivity in a similar fashion to bariatric surgery [89]. This is accompanied by profound increases in unconjugated bile acid levels after 6 months, similar to bariatric surgery, although a temporal dissociation between bile acids, GLP-1 and FGF19 requires caution about causality. These effects were accompanied by increased abundance of typical small intestinal bacteria such as Proteobacteria, Veillonella, and Lactobacillus spp. in feces [90].

## 5. Conclusions

In this paper we aimed to describe the reciprocal interaction between the gut microbiome and bile acids from a general and clinical point of view (see Figure 1). Clearly, this relationship is complex and interaction between the intestinal gut microbiome and bile acids is not well understood. Nevertheless, our gut microbiome and bile acids are part of a well-orchestrated and maintained gut–liver axis interaction, which is quite homeostatic under normal circumstances. However, dysbiosis is around the corner under different pathological circumstances including obesity, T2DM, Western lifestyle and antibiotic treatment. It is safe to say that we could not be extensive and complete in our literature analysis due to the abundance of studies on gut bacteria and their metabolites, which may be a limitation of this paper. Additionally, we discussed different types of studies with respect to design, study participants, follow-up and samples. Notably, some studies investigated fecal bile acids, whereas others had an interest in plasma bile acids. It should be realized that the real interesting biology of the gut microbiome and bile acids takes place in the enterohepatic circulation, which includes portal vein sampling and liver biopsies. Not surprisingly, due to ethical constraints and inaccessibility of non-invasive sampling, the true interaction of the gut microbiota within our enterohepatic circulation will remain a black box for the next years. Nevertheless, with the exception of the RYGB it can be concluded that a higher gut microbiome diversity is associated with lower, and in particular secondary, bile acids. Future studies should focus on the effect of nutrition to regain and retain a qualitative healthy gut microbiome composition from a general point of view. The heterogeneity of studies, diseases, bacterial species and (epi)genetic influences such as nutrition may challenge specific and detailed interventions that aim to tackle the gut microbiome and bile acids.

## Figures and Tables

**Figure 1 ijms-24-01816-f001:**
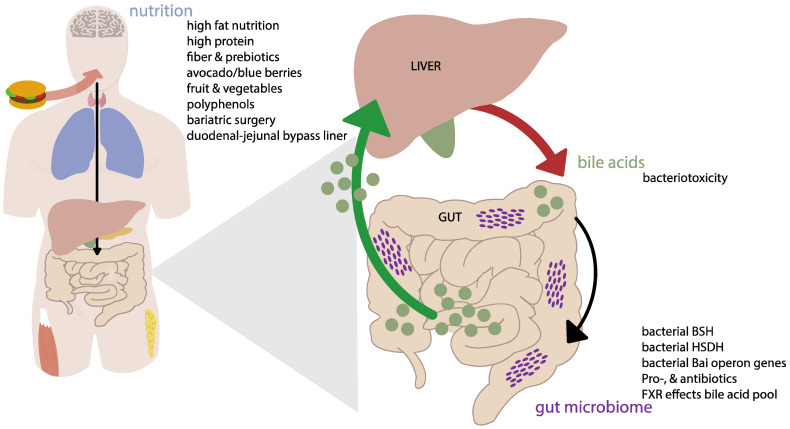
Conceptual scheme of the interaction between the microbiome, bile acids and nutrition. Mechanisms are described in detail in the text. Abbreviations: BSH, bile salt hydrolase; HSDH, hydroxysteroid dehydrogenase; Bai, bile acid inducible; FXR, Farnesoid X receptor.

## Data Availability

Not applicable.

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
