# Peer review of "The Black Box Orchestra of Gut Bacteria and Bile Acids: Who Is the Conductor?"

_ijms, 2023, doi:10.3390/ijms24031816_

Round 1

Reviewer 1 Report

This review highlights the current understanding of the interactions between the gut microbiome and bile acid transformations and addresses future directions to better understand these complex associations. It is an exciting topic that deserves more attention. It has a lovely title. The review must emphasize how microbial communities in a healthy and dysbiotic individual may or may not affect bile acid levels and potentially mitigate bile acid synthesis and disease onset. 

A discussion of bile acid metabolism and pathways should also be included. As mentioned above, authors need to provide more clarity about the role of bile acids in regulating gut microbial composition. 

Minor mistakes : 

Line 12: T2DM : Define 

Line 13-15: repeated below in the introduction 

Line 17-19: Bai operon genes encode enzymes. Revise accordingly. 

Line 35: In this current review paper:: Revise. 

Section: 1.1: Poor referencing. 

Line 39: cite reference 

Line 46: amphipathic steroids: > amphipathic acidic steroids

Section 1.2. Poorly referenced. 

Section 1.3. The microbiome part needs referencing and should be more detailed. 

Line 116: Repeated in the abstract. 

Line 123-123: Revise 

Line 123: ... Islam et al. correct and provide a reference. 

Line 139 onwords, needs supporting references. 

Line 148: Check bacteria names: like _Also "Bacteria" >B should not be in capital letters. 

Line 156: Bacteria name should be in italics front throughout the text. e.g>Faecaliacterium prausnitzii, and Line 157 correct as > F. prausnitzii. 

Line 176-181: Cite supporting references.

Section 3. Cite supporting references.

Section 3.1: Cite supporting references.

Line 195: was show > was shown 

Line 201: Reference 

Line 204: Reference 

3.3. Revise 

Section 3.4: 

Line 220: what do you mean: ....or applied to the body? 

Line: line 220-221: we have yeast probiotics also. 

Line 223: Anaerobutyricum soehngenni > italic font 

Line 226: Avoid using terms like : Indeed

Line 229-234:/Line 246-247: name of each bacterium should be written in italic font. 

Author Response

Reviewer 1

We thank reviewer 1 for the valuable remarks regarding the manuscript. We tried to adapt the manuscript as much as possible. However, we are not sure that we fully succeeded in doing so because we found it difficult to judge whether some of the points were just general remarks or pinpointed suggestions to be adapted. If needed we would welcome specific suggestions.

Because of the nature of this short and concise review, we did not take the liberty to fully expand on the different topics, but provided an overview.

  1. This review highlights the current understanding of the interactions between the gut microbiome and bile acid transformations and addresses future directions to better understand these complex associations. It is an exciting topic that deserves more attention. It has a lovely title. The review must emphasize how microbial communities in a healthy and dysbiotic individual may or may not affect bile acid levels and potentially mitigate bile acid synthesis and disease onset. A discussion of bile acid metabolism and pathways should also be included. As mentioned above, authors need to provide more clarity about the role of bile acids in regulating gut microbial composition. 

Authors: See above. We found it difficult to judge whether some of the points were just general remarks or pinpointed suggestions to be adapted. If needed we would welcome specific suggestions (also with regard to the role of bile acids in regulating gut microbial composition).

Minor mistakes:

  1. Line 12: T2DM : Define 

Authors: We have left out the abbreviation.

  1. Line 13-15: repeated below in the introduction 

Authors: We agree that we used the sentences also in the introduction, but it is an important point therefore we also want to mention it in the abstract.

  1. Line 17-19: Bai operon genes encode enzymes. Revise accordingly. 

Authors: We have corrected it to enzymes encoded in the bile acid inducible (Bai) operon genes.

  1. Line 35: In this current review paper:: Revise. 

Authors: We do not know what the reviewer means exactly: what should we revise?

  1. Section: 1.1: Poor referencing. 

Authors: We added references 3, 8-10.

  1. Line 39: cite reference 

Authors: We added reference 2.

  1. Line 46: amphipathic steroids: > amphipathic acidic steroids

Authors: We corrected amphipathic steroids to amphipathic acidic steroids

  1. Section 1.2.Poorly referenced. 

Authors: We added references 11-15.

  1. Section 1.3. The microbiome part needs referencing and should be more detailed. 

Authors: We understand the point of the reviewer. However, here (since this is a concise review) we briefly mention the basic background of bile acid metabolism here and focus more on detail in later sections. Can the reviewer agree on this? We added reference 8.

  1. Line 116: Repeated in the abstract. 

Authors: We acknowledge this repetition. The abstract contains the most important ideas/concepts of the paper, hence repetition may occur. We shortened the sentence in the abstract

  1. Line 123-123: Revise 

Authors: Here is replaced with moreover

  1. Line 123: ... Islam et al. correct and provide a reference. 

Authors: We added reference 28

  1. Line 139 onwords, needs supporting references. 

Authors: We added references 8 and 28

  1. Line 148: Check bacteria names: like _Also "Bacteria" >B should not be in capital letters. 

Authors: We corrected it.

  1. Line 156: Bacteria name should be in italics front throughout the text. e.g>Faecaliacterium prausnitzii, and Line 157 correct as > F. prausnitzii. 

Authors: We corrected it.

  1. Line 176-181: Cite supporting references.

Authors: We added references 10,11,17-20 and 28

  1. Section 3. Cite supporting references.

Authors: We added references 44 and 45

  1. Section 3.1: Cite supporting references

Authors: We added references 46 and 47

  1. Line 195: was show > was shown 

Authors: We corrected it.

  1. Line 201: Reference 

Authors: We added reference 44

  1. Line 204: Reference 

Authors: We added reference 44

  1. 3. Revise 

Authors: We corrected it for enzymes encoded in the bile acid inducible (Bai) operon genes

  1. Line 220: what do you mean: ....or applied to the body? 

Authors: We mean that probiotics can be either consumed as supplements or added to food. We removed the part of the sentence ‘applied to the body’ in order to avoid misunderstanding

  1. Line: line 220-221: we have yeast probiotics also. 

Authors: We are aware of the fact that priobiotics come in various forms, however for this review we only focus on bacterial strains.

  1. Line 223: Anaerobutyricum soehngenni > italic font 

Authors: We corrected it

  1. Line 226:Avoid using terms like : Indeed

Authors: We removed the term indeed

  1. Line 229-234:/Line 246-247: name of each bacterium should be written in italic font. 

Authors: We corrected it

Reviewer 2 Report

The review article entitled "The black box orchestra of gut bacteria and bile acids: who is the conductor?" by S. Majait et al. is interesting, but there are some flaws that need to be addressed.
Please reconsider the title of this article. The author made many claims in the text but did not provide references, for example lines 38–42, 55–57, and more.

The author needs to rewrite the introduction part in a coherent manner, with references where needed.

The author mentioned bile acid deconjugation and dihydroxylation in line 64, but I couldn't find it. The author should point out which figure is relevant to which heading.

The authors claimed that conjugated bile acid is toxic and deconjugated is more toxic; which form of bile acid is involved in the increased levels of T2DM, conjugated or unconjugated? Second, the author should provide enough evidence to defend these clams; otherwise, please remove these clams.

Please make the article more coherent, proofread it for errors and grammar, and take the services of a native English speaker if needed .

Author Response

Reviewer 2

We thank reviewer 2 for the valuable remarks regarding the manuscript.

  1. The review article entitled "The black box orchestra of gut bacteria and bile acids: who is the conductor?" by S. Majait et al. is interesting, but there are some flaws that need to be addressed.
    Please reconsider the title of this article. The author made many claims in the text but did not provide references, for example lines 38–42, 55–57, and more.

Authors: We understand the remarks of the reviewer with regards to the title. We aimed to provide a more or less provocative title which we explain in the conclusion (line 312-315). We prefer to leave the title as it is if the reviewer can agree with this.

With respect to the references we added these were requested.

  1. The author needs to rewrite the introduction part in a coherent manner, with references where needed.

Authors: We have tried to clarify the writing, however we are not certain what level of coherence the reviewer refers to

  1. The author mentioned bile acid deconjugation and dihydroxylation in line 64, but I couldn't find it. The author should point out which figure is relevant to which heading.

Authors: Figure was removed but the text was not adapted to it. We corrected it.

  1. The authors claimed that conjugated bile acid is toxic and deconjugated is more toxic; which form of bile acid is involved in the increased levels of T2DM, conjugated or unconjugated? Second, the author should provide enough evidence to defend these clams; otherwise, please remove these clams.

Authors: We do not claim that conjugated bile acid is toxic and deconjugated is more toxic. We mention that conjugation is known to influence the toxicity of bile acids. We have added reference 8.

  1. Please make the article more coherent, proofread it for errors and grammar, and take the services of a native English speaker if needed.

Authors: We checked the paper.

Round 2

Reviewer 1 Report

The paper can be accepted in its current form as a mini-review. 

Reviewer 2 Report

The addressed responses were satisfying.